# Conformational Stability and Denaturation Processes of Proteins Investigated by Electrophoresis under Extreme Conditions

**DOI:** 10.3390/molecules27206861

**Published:** 2022-10-13

**Authors:** Patrick Masson, Sofya Lushchekina

**Affiliations:** 1Biochemical Neuropharmacology Laboratory, Kazan Federal University, Kremlievskaya Str. 18, 420111 Kazan, Russia; 2Emanuel Institute of Biochemical Physics, Russian Academy of Sciences, Kosygin Str. 4, 119334 Moscow, Russia

**Keywords:** gel electrophoresis, capillary electrophoresis, protein denaturation, unfolding, refolding, stability

## Abstract

The functional structure of proteins results from marginally stable folded conformations. Reversible unfolding, irreversible denaturation, and deterioration can be caused by chemical and physical agents due to changes in the physicochemical conditions of pH, ionic strength, temperature, pressure, and electric field or due to the presence of a cosolvent that perturbs the delicate balance between stabilizing and destabilizing interactions and eventually induces chemical modifications. For most proteins, denaturation is a complex process involving transient intermediates in several reversible and eventually irreversible steps. Knowledge of protein stability and denaturation processes is mandatory for the development of enzymes as industrial catalysts, biopharmaceuticals, analytical and medical bioreagents, and safe industrial food. Electrophoresis techniques operating under extreme conditions are convenient tools for analyzing unfolding transitions, trapping transient intermediates, and gaining insight into the mechanisms of denaturation processes. Moreover, quantitative analysis of electrophoretic mobility transition curves allows the estimation of the conformational stability of proteins. These approaches include polyacrylamide gel electrophoresis and capillary zone electrophoresis under cold, heat, and hydrostatic pressure and in the presence of non-ionic denaturing agents or stabilizers such as polyols and heavy water. Lastly, after exposure to extremes of physical conditions, electrophoresis under standard conditions provides information on irreversible processes, slow conformational drifts, and slow renaturation processes. The impressive developments of enzyme technology with multiple applications in fine chemistry, biopharmaceutics, and nanomedicine prompted us to revisit the potentialities of these electrophoretic approaches. This feature review is illustrated with published and unpublished results obtained by the authors on cholinesterases and paraoxonase, two physiologically and toxicologically important enzymes.

## 1. Introduction

The study of protein unfolding/denaturation provides a considerable mass of information on stability, conformational dynamics, folding/unfolding intermediates, post-denaturation/renaturation events, and irreversible covalent modifications. Among the approaches to investigate protein stability and denaturation processes, electrophoretic methods under denaturing conditions have attracted scientists due to their simplicity and high sensitivity. These methods, in general accessible, provide visual descriptions of unfolding transitions and may lead to quantitative information, in particular on energetic and volumetric changes accompanying transitions, and about the formation of transient intermediates.

Although polyacrylamide gel electrophoresis (PAGE) in the presence of dodecyl sulfate (SDS) is certainly the most popular electrophoretic technique under denaturing conditions, we will not talk about this. It is mostly used for the purity control of protein preparations and the determination of protein size and number of subunits covalently (disulfide-bonded) or non-covalently bound in quaternary structures. These important applications have been extensively reviewed since the introduction of this method [1], and new developments of capillary zone electrophoresis (CZE) in the presence of SDS are still in progress [2,3]. The use of CZE to investigate the stability of proteins, in particular proteins of pharmaceutical interest, has been reviewed [4,5]. However, most of the pioneering works were performed using PAGE techniques. Thus, in this review, we mostly focus on experimental classical gel electrophoresis approaches that have been developed to describe unfolding/refolding processes of proteins, irreversible inactivation of enzymes, and protein denaturation and to estimate parameters of protein conformational stability.

## 2. Structural and Conformational Stability of Proteins

The functional activity of proteins mostly depends on the integrity of the three-dimensional (3D) structure. The tertiary (3D) structure of proteins results from the folding and stabilization of the polypeptide chain. Higher structural and functional organization levels lead to quaternary and quinary structures. These supramolecular levels may result from non-covalent binding and/or covalent bonding of homologous and/or heterologous subunits. Covalent binding can be disulfide bridges between cysteine residues and other post-translational modifications on amino acid side chains (e.g., transglutaminase-mediated isopeptide cross-links [6]) or on glycan units in the case of glycoproteins. A remarkable example of such complex multi-subunit structures is represented by cholinesterases (ChEs) acetylcholinesterase (AChE) and butyrylcholinesterase (BChE) (Figure 1), key enzymes of the cholinergic system that terminate the action of acetylcholine in the central nervous system and at neuromuscular junctions [7]. In addition, BChE is of toxicological and pharmacological importance in scavenging or hydrolyzing toxic esters [8].

Engineered covalent modifications of proteins such as “capping” with polyethylene glycol, polysialic acid, or other polymeric shells and encapsulation into nanocontainers have proved to considerably increase the stability of pharmaceutical proteins [12], to increase their residence time in blood after injection, and to prevent adverse immune responses to injected free or nanoencapsulated heterologous proteins [13,14].

## 3. Protein Stability, Unfolding, Denaturation, and Deterioration

The functional 3D structure of proteins results from a delicate balance between a paramount number of elementary weak interactions. The physicochemical conditions of the environment, mostly hydration, pH, temperature (*T*), pressure (*P*), and salinity, play a determining role in maintaining the native (*N*) folded conformation.

In the past century, starting in the 1930s with Hsien Wu [15], protein biophysicist pioneers provided a considerable mass of information on the structure and conformational dynamics of proteins and about structural and energetic changes associated with folding/unfolding processes, including irreversible denaturation [16,17,18,19,20,21]. These pioneering works have had a considerable impact on protein engineering, food biotechnology, the development of stable bio(nano)pharmaceuticals [22,23], and the understanding and treatment of certain genetic and degenerative diseases, including protein misfolding diseases [24]. Indeed, the research on extremozymes from organisms living in extreme biotopes and the creation of novel enzymes capable of working under non-conventional conditions, i.e., at high temperature and high pressure in water-restricted media, as industrial catalysts for fine chemical synthesis and as biological reagents for clinical chemistry are among the main goals of protein engineering. The success of polymerase chain reaction (PCR) tests for the detection of genetic defects and diagnosis of infectious diseases is one of the most popular achievements of thermophilic enzyme biotechnology.

The native conformation of proteins is thermodynamically the most stable under optimum physicochemical conditions. Under these conditions, the free energy change (Δ*G*) of the protein–solvent system as a function of the different variables is minimum:(1)$$\Delta G=\;\Delta H-T({\Delta S}_{\mathrm{solv}}+\Delta S_{\mathrm{conf}})+P\Delta V.$$

In Equation (1), Δ*H* is the enthalpy change resulting from the multiple intramolecular interactions, *T* is the absolute temperature, Δ*S*_solv_ and Δ*S*_conf_ are the solvation entropy and the conformation entropy, *P* is the hydrostatic pressure, and Δ*V* is the volume change associated with the folding of the polypeptide chain.

The free energy of protein stabilization, Δ*G*, is the sum of partial free energy of numerous (*n*) non-covalent (*el*, electrostatic; *h*, hydrophobic; *vw*, van der Waals; conf, conformational; *H*, hydrogen; *int*, internal apolar interactions) and covalent contributions (mostly disulfide bridges, *ss*) (Equation (2)):(2)$$\Delta G=\sum_{1}^{n}(\Delta G_{i,el}+\Delta G_{j,h}+\Delta G_{k,vw}+\Delta G_{l,conf}+\Delta G_{m,H})+\;\Delta G_{int}+\;\Delta G_{SS}.$$

The difference in free energy between protein native and denatured states is low, of the order of 25–60 kJ·mol^−1^ around physiological temperatures [25]. On the contrary, enthalpic and entropic changes accompanying denaturation are large, ranging from 500 to 5000 kJ·mol^−1^. The small net energy for stabilization of proteins corresponds approximately to the free energy needed to stabilize one or two hydrogen bonds (Δ*G_stabil,H_* ≤ −29 kJ·mol^−1^). As a result, proteins are marginally stable, and their stability zone is highly dependent on the physicochemical conditions of the medium. The highly positive conformational entropy dominating in the denatured state is compensated in the native state by enthalpy and solvation entropy. The latter is minimized by decreased hydrophobic areas in contact with the solvent (water).

## 4. Phenomenology of Protein Denaturation

Protein denaturation corresponds to the transition between the ordered and functional native state (*N*) of minimum free energy to a disordered state (*U*) of higher energy. When this transition is complete, the polypeptide chain is called a “statistic coil”, and all segments (i.e., amino acids) are solvated. Partially folded intermediate states (*I*) may exist or transiently appear. All forms may be in rapid or slow equilibrium. The transition may be reversible or not. In the latter case, unfolded chain leads to denatured protein (*D*). Irreversible denaturation may be the result of chemical deteriorations (side chain reactions, cross-links, partial hydrolysis, and other bond cleavages) or the formation of scrambled unfolded structures.

Unfolding can be induced by changing the physicochemical conditions of the medium, including *T*, *P*, pH, ionic strength, dielectric constant, electric field, ultrasound irradiation, ionizing radiation, and chemical agents competing with intramolecular non-covalent bonds such as hydrogen bonds (e.g., organic solvents, urea, guanidinium, SDS). This may also lead to increased stability of *N* states. For example, replacement of water (H_2_O) by heavy water (D_2_O) as a solvent increases the stability of proteins under pressure because D-bonds are stronger than H-bonds (Figure 2A), or stabilizing cosolvents and molecules such as polyols, e.g., trehalose, that substitute to water can be added (Figure 2B). On the other hand, the presence of a destabilizing cosolvent makes proteins more sensitive to the denaturing action of pressure (Figure 2B).

Unfolding of small proteins is in general a fast and highly cooperative transition between *N* and *U* states without detectable intermediates.
(3)$$N\begin{array}{c} k_{1}\\ ⇌\\ k_{-1} \end{array}U,$$
where *k*_1_ and *k*_−1_ are the kinetic constants of unfolding and refolding, respectively, and *K_U_* is the transition equilibrium constant.
(4)$$K_{U}=\frac{k_{1}}{k_{-1}}=\frac{\left[U\right]}{\left[N\right]}=e^{-\Delta G_{U}/RT}.$$

As for single molecule transitions, the rate (*k*) of interconversion between the two conformational states *N* and *U* depends only on the properties of both states and characteristics of the activated native state, *N**. The number of molecules in the *N** state is small. Thus, the *N** state represents a very small fraction of the total concentration.

For the transition N→U, the rate *k* is proportional to the equilibrium constant, *K** = [*N**]/[*N*], that measures the probability to jump to the activated state:(5)$$k=\frac{k_{B}T}{h}K^{*},$$
where *k_B_* is the Boltzmann constant and *h* is the Planck constant.

Proteins composed of several domains and oligomeric proteins are heterogeneous thermodynamic systems, and thus, their unfolding process is more complex, involving consecutive intermediate states (*I_n_*) more or less stable.
(6)$$N⇌\left(I_{1}\ldots ⇌\ldots I_{n}\right)\;⇌\;U.$$

Irreversibility of unfolding transitions leads to denatured states (*D*).
(7)N⇌U →D

Several mechanisms account for irreversible denaturation [27]. Irreversible denaturation can be due to the formation of scrambled partially unfolded conformations and subsequent gelation, but in general irreversible denaturation results from covalent bond breakages or the formation of new bonds affecting the polypeptide chain (hydrolysis of amide bonds), side-chain modifications leading to intramolecular bridges, thiol/disulfide exchanges, cross-links with glycan components (e.g., Maillard reaction), etc. [28]. These pH-dependent reactions are favored by heat.

## 5. Analysis of Reversible Denaturation Processes

The kinetics and equilibrium of protein unfolding can be investigated by following the loss of enzyme activity or binding affinity and monitoring the changes in physicochemical and structural parameters as a function of external thermodynamic variables (*T*, *P*), pH, or concentration of the denaturing agent. Typical transition curves in conditions of reversibility are sigmoids (Figure 3A).

Numerous techniques are available for investigating the reversible and irreversible unfolding and denaturation of proteins. Methodological principles have been described in numerous reviews and books (see for example [18,19,29,30,31,32]). Most spectrometric methods (UV and visible spectrophotometry, fluorescence, infrared, Raman, NMR, mass spectrometry, circular dichroism, light scattering, in particular small-angle X-ray scattering, neutron scattering), as well as hydrodynamic methods (viscometry, sedimentation, light scattering, size exclusion chromatography, and electrophoresis), differential scanning calorimetry (DSC), and techniques allowing the investigation of functional properties (enzymatic activity, ligand binding, immunology), can be used. Kinetic studies are in general performed with spectroscopic techniques that offer the possibility to detect fast changes (down to 10^−9^ s by fluorescence measurements). Classical hydrodynamic techniques do not allow the monitoring of fast changes, and they are mostly used in equilibrium studies. They provide information on shape and volume changes, dissociation of oligomeric proteins, and aggregation of (fully and partially) unfolded states. Hydrodynamic techniques allow separating, in the transition zone, intermediates and species of different sizes and hydrodynamic volumes. The analysis of denaturation profiles, then, provides information on the mechanism of transition and the number of intermediates. Spectroscopic techniques allow easy access to the energetics of transitions, but the method of choice for determining energy changes and reversibility of transition is DSC, which provides thermodynamic quantities without making hypotheses on the transition mechanisms. DSC was initially developed for the study of reversible heat-induced unfolding transitions and was subsequently refined for the study of cold denaturation [33,34,35,36] and irreversible temperature-induced denaturation transitions [27].

The simple analysis of transition curves is based on the hypothesis that unfolding is a transition between two states, *N* and *U*. This statement is in general correct for small proteins. For large proteins, as we already said, there are multiple intermediates. However, the simple two-state analysis can be used and leads to apparent parameters of phenomenological interest.

Considering *y* as the experimentally observed parameter for viewing the transition *N U*, with *y_N_* and *y_U_*, the values of *y* in *N* and *U* states, respectively, for each point of the transition,
(8)$$y=f_{N}y_{N}+f_{U}y_{U},$$
where *f_N_* and *f_U_* are the fractions of protein in the *N* and *U* states, respectively. Because *f_N_* + *f_U_* = 1,
(9)$$y=y_{N}+f_{U}\left(y_{U}-y_{N}\right),$$
and
(10)$$f_{U}=\frac{y-y_{N}}{y_{U}-y_{U}}.$$

Each measure of *y* gives a unique value of *f_U_*. Then, from the unfolded fraction, the equilibrium constant *K_U_* and thus the free energy change Δ*G_U_* can be determined for each point of the transition:(11)$$K_{U}=\frac{f_{U}}{1-f_{U}}=\frac{y_{N-}y}{y-y_{U}},$$
with
(12)$$\Delta G_{U}^{0}=-RT\ln K_{U}.$$

At the transition midpoint where *K_U_* = 1, Δ*G_U_* = 0 (Figure 3B).

The shape of the transition curve provides information on the mechanism of unfolding. Sharp transitions in general fit with the two-state model, rather than smooth sigmoidal transitions that may indicate multiple intermediates. Moreover, if the transition shows several steps, it signs the existence of stable intermediates. The simplest test to prove that unfolding is a two-state or a multi-state transition is to monitor the changes of different *y* parameters as a function of the denaturation variable. If transition curves coincide, then the transition obeys the two-state model. If transition curves are shifted and/or show different slopes, then the transition is stepwise (Figure 3B).

The functional property parameters, e.g., enzyme activity, ligand binding, and antibody recognition, are the most sensitive as they reflect both topography of specific sites and protein molecular dynamics. Functionality rapidly decreases as the variable intensity increases. Then, two other parameters can be considered, parameters that measure the change in tertiary structure (tryptophan fluorescence, binding of specific probes for thiols (free cysteines) and hydrophobic patches, proton/deuteron (H/D) exchange) and in secondary structure (UV absorbance of aromatic residues, circular dichroism in far UV, infrared and Raman spectroscopy, neutron scattering).

## 6. Determination of Protein Stability

Thermodynamic parameters that describe reversible unfolding processes (cf. Equation (1)) are most often determined from equilibrium constants corresponding to transitions N⇌U or from rates of exchange H/D. However, for low- and high-temperature-induced unfolding, they can be directly determined from microcalorimetric measurements by using DSC.

The free energy change in water (w) in the absence of a denaturing agent, ${\Delta G}_{U}^{w}$, is a measure of the net stability of proteins:(13)$${\Delta G}_{\mathrm{stab}}^{w}=-{\Delta G}_{U}^{w}$$

For most proteins, ${\Delta G}_{\mathrm{stab}}^{w}$ ranges between −20 and −70 kJ·mol^−1^. Thus, the maximum equilibrium constants, *K*_N_ = [*N*]/[*U*], range between 10^3^ and 10^11^. Therefore, the probability of spontaneous denaturation under optimal physicochemical conditions is between 10^−3^ and 10^−11^.

### 6.1. Experimental Techniques

Experimental determination of ${\Delta G}_{U}^{w}$ from denaturation curves at pH and constant temperature, using a denaturing agent such as urea or guanidinium chloride, is popular. The denaturing agent induces a complete unfolding of proteins, competing with water for the formation of H-bonds with polypeptide -CO- and -NH- bonds, and solvated amino acid side chains. The mechanism of unfolding is in general less complex than for heat denaturation and other physical or chemical agents.

Different approaches allow calculating ${\Delta G}_{U}^{w}$ by extrapolating *K*_U_ at concentration 0 of the denaturing agent [37,38,39]. The simplest method is the linear extrapolation method (LEM). It is based on the observation that in the transition zone, Δ*G*_U_ change is linear with the concentration of the denaturing agent [*d*]. It is hypothesized that it is still true when [*d*]→0. Then,
(14)$$\Delta G_{U}=\;\Delta G_{U}^{w}-m\left[d\right].$$

At the transition midpoint, [*d*]_0.5_:(15)$${\left(\frac{d\Delta G_{U}}{d\Delta d}\right)}_{{\left[d\right]}_{0.5}}=m.$$

The transition midpoint depends on both ${\Delta G}_{U}^{w}$ and *m*. The parameter *m* (kJ·mol^−1^·M^−1^) measures the slope of the transition; it expresses the dependence of Δ*G*_U_ on the concentration of the denaturing agent. Thus, *m* reflects the denaturing power of a chemical compound. It is correlated with the change in solvent-accessible surface area (SASA) of the protein that unfolds [40]. This analysis revealed that solvent-denatured proteins still contain some residual structure [41]. Lastly, the validity of LEM implies that *m* is independent of the concentration of the denaturing agent. A recent study demonstrated that due to compensatory effects this assumption is valid with urea but may provide erroneous results in the case of other denaturing agents [42].

The parameters [*d*]_0.5_ and *m* also allow a qualitative comparison of the stability of engineered proteins (mutants and chemically modified forms). Indeed, mutations and chemical modifications cause generally small changes in free energy of denaturation:(16)δG=δH−δ(TΔS)

This induces a shift in transition midpoint and a change in cooperativity of the unfolding transition.

The determination of thermodynamic parameters of stability is possible as a function of temperature (heat and cold) and hydrostatic pressure. As mentioned above, the presence of a protecting or a denaturing agent or a cosolvent in the medium allows shifting the unfolding transitions toward stabilization or destabilization of folded structures, causing ${\Delta G}_{U}^{w}$ and shifts in the unfolding transition midpoint (cf. Figure 2B).

The change in Δ*G*_U_ with pressure and temperature is described by the Hawley master equation (Equation (17)) [43], which has been the subject of numerous investigations [44,45]:(17)$$\Delta G=\frac{\Delta \beta }{2}{\left(P-P_{0}\right)}^{2}+\Delta \alpha \left(P-P_{0}\right)\left(T-T_{0}\right)-\frac{\Delta C_{p}}{2T_{0}}{\left(T-T_{0}\right)}^{2}\;+\Delta V_{0}\left(P-P_{0}\right)\Delta S_{0\;}\left(T-T_{0}\right)+\Delta G_{0},$$
where *β* is the compressibility factor $\left(\beta ={\left(\frac{\partial V}{\partial P}\right)}_{T}\right)$, *α* is the thermal expansivity factor $\left(\alpha ={\left(\frac{\partial V}{\partial T}\right)}_{P}\right)$, and *C_p_* is the heat capacity $\left(C_{P}=T{\left(\frac{\partial S}{\partial T}\right)}_{P}\right)$. Figure 4 shows the change in Δ*G*_U_ of a small protein as a function of pressure and temperature as described by Equation (17).

Determination of the equilibrium constant *K*_U_ as a function of temperature leads, from van’t Hoff plots, to enthalpy change, Δ*H*_U_, accompanying unfolding,
(18)$$\frac{\partial \ln K_{U}}{\partial \left(1/T\right)}=-\frac{\Delta H_{U}}{R}.$$

The presence of a denaturing (or a stabilizing (protecting)) agent (*a*) in the medium shifts the thermal transition towards lower or higher temperatures. Considering the change in *K*_U_ as a function of *T* and [*a*], it follows that
(19)$$d\ln K_{U}={\left(\frac{\partial \ln K_{U}}{\partial T}\right)}_{\left[a\right]}dT+{\left(\frac{\partial \ln K_{U}}{\partial \left[a\right]}\right)}_{T}d\ln\left[a\right]$$

The effect of *a* on the temperature transition midpoint, *T*_0.5_, is as follows [18]:(20)$$\frac{dT_{0.5}}{d\ln\left[a\right]}=-\frac{\nu RT_{0.5}}{\Delta H_{U}}.$$

In Equation (20), ν is the interaction factor between the protein and the denaturing/protecting agent *a* (cf. Equations (14) and (15)):(21)$$K_{U}=A{\left[a\right]}^{\nu }\;\mathrm{and}\;\frac{\partial \ln K_{U}}{\partial \ln\left[a\right]}=\nu .$$

The analysis of temperature-induced unfolding theoretically allows the determination of the entropy of denaturation:(22)$$\Delta S_{U}=\frac{\Delta H_{U}-\Delta G_{U}}{T}.$$

At the transition midpoint, where Δ*G*_U_ = 0,
(23)$$\Delta S_{U}=\Delta H_{U}/T_{0.5}$$

Then,
(24)$$\Delta G_{U}=\Delta H_{U}\left(1-\frac{T}{T_{0.5}}\right).$$

In fact, it is experimentally observed that Δ*H*_U_ and Δ*S*_U_ are strongly dependent on *T.* In particular, Δ*H*_U_ is negative at low temperatures and positive at high temperatures [17].

The change in Δ*H*_U_ with *T* (Kirchoff equation, Equation (25)) is the heat capacity change Δ*C_p,_*_U_, the difference in heat capacity between N and U states, i.e., the heat absorbed during the unfolding process (cf. Equation (17)).
(25)$$\frac{\partial \Delta H_{U}}{\partial T}=\Delta C_{p,U}.$$

The heat capacity change of proteins is currently determined by using DSC. All proteins between 20 and 30 °C regardless of the pH show similar heat capacity, 1.33 ± 0.08 kJ × K^−1^ × kg^−1^. Δ*C_p_,*_U_ is highly positive and constant up to 80 °C, ranging between 4 and 12 kJ × K^−1^ [46]; it depends on protein hydrophobicity. Thus, the enthalpy change accompanying the thermal unfolding transition, Δ*H*_U_, reflects changes in hydrophobic interactions between the solvent and apolar groups initially buried in the protein core. This change reflects the denaturing effect of heat due to the solvation of internal hydrophobic groups.

Cold temperatures may induce the dissociation of oligomers and the unfolding of certain proteins [33,34,47]. Cold-induced unfolding transitions are fully reversible and may show hysteresis during renaturation.

High hydrostatic pressure (*P* > 1.5 kbar, i.e., >150 MPa) induces dissociation of oligomeric structures and unfolding. As mentioned, the transition is dependent on other physicochemical variables (e.g., *T*, pH, salts, presence of a cosolvent) (cf. Figure 2A,B and Figure 4). The denaturing effect of pressure mostly results from the breakage of hydrophobic bonds and hydrophobic solvation. In general, pressure denaturation is less extensive than denaturation induced by denaturing agents or heat [48]. Between 1 kbar and 2 kbar (100–200 MPa), unfolding is accompanied by an increase in volume. Beyond 200 MPa, important negative volume changes (Δ*V*) are observed [29,48,49,50]. These Δ*V* < 0 values are correlated with solvation entropy resulting from the hydrophobic effect. Volume changes accompanying the unfolding process can be determined by measuring the change in *K*_U_ as a function of pressure (*P*):(26)$${\left(\frac{\partial \ln K_{U}}{\partial P}\right)}_{T}=-\Delta V/RT$$
or
(27)$${\left(\frac{\partial \Delta G_{U}}{\partial P}\right)}_{T}=\Delta V.$$

Thus, Δ*V* is an important thermodynamic parameter. It is also a mechanistic criterion whose sign and magnitude reflect the extent and nature of structural changes accompanying the transition. Δ*V* is a composite parameter, the sum of different volumetric contributions (Equation (28)):(28)$$\Delta V=\sum {\Delta V}_{\mathrm{int}}+{\Delta V}_{\mathrm{conf}}+{\Delta V}_{w}.$$

These are interactions between chemical groups (Δ*V*_int_), conformational terms (Δ*V*_conf_), and changes in solvation (Δ*V*_w_) of the system’s various components, including changes in the solvent structure. Because a change in hydration of amino acids occurs during the unfolding process (H-bonding of water with polar groups, hydrophobic solvation, electrostriction of water around charged groups), the Δ*V*_w_ contribution is large.

### 6.2. Computational Approaches

Stability can also be estimated by using computational methods. Initial predictive methods for secondary and tertiary structures from primary sequences provided information about energy for stabilizing α and β structures. However, the free energy of stabilization of 3D structure is only a very small part of the total free energy of proteins, and calculations of ${\Delta G}_{\mathrm{stab}}^{w}$ must take into account the interaction of proteins with their environment. Thus, these methods lead to large errors. With the development of supercomputing facilities, starting from 3D structure, modern all-atom and coarse-grained molecular dynamic (MD) simulations became powerful tools for assessing protein stability and improving the functionality of protein mutants [9,51,52,53,54].

MD simulations require knowledge of the initial 3D structure of proteins, preferably obtained by X-ray crystallography and/or NMR. Otherwise, in the absence of experimental structural information about the overall protein structure or a part of the structure, homology modeling can be performed [55,56,57]. Very recently, artificial intelligence programs such as AlphaFold2 [58] have been used to provide protein structure from a sequence. Analysis of unconstrained and unbiased classical MD trajectories from several nanoseconds up to microseconds long for wild-type proteins, compared to their natural mutants and genetically engineered variants, is a quite straightforward way to assess protein stability and the impact of sequence modification on this parameter. Evolution of distances between certain atoms or groups of atoms (angles, dihedrals, etc.) along MD trajectory allows the control of the stability of active sites and catalytic groups and, thus, the control of enzyme functionality (examples of such an analysis for BChE natural variants can be found in [59,60,61,62]). Protein stability along an MD trajectory is characterized through simple metrics calculated directly from protein atomic coordinates at each recorded timestep of the trajectory: root mean square deviation (RMSD), root mean square fluctuation (RMSF) of individual residues or other protein fragments, radius of gyration of protein (R_g_), SASA, and hydration (number of water molecules in certain area near the protein). These parameters can be compared and correlated to available experimental data such as mean square deviation derived from neutron scattering data, the molecular radius R, and denaturation parameters.

Overall analysis of MD trajectories revealing conformational transitions in a protein and the mobility of its segments includes such methods as principal component analysis (PCA) [63], dynamic residue network analysis [64], dynamic cross-correlation (DCC) analysis [65], normal mode analysis (NMA) [66], and Markov state models (MSMs) [67,68]. A few recent examples of the application of these approaches of analysis of MD trajectories providing useful information on stability, the effect of mutations, and allosteric effects of various proteins, including those of SARS-CoV-2 [69], can be found in [70,71,72,73,74].

For quantifying energetics of protein stability and interactions, there are numerous methods for free energy estimation through constrained or otherwise biased MD simulations. These are steered molecular dynamics (SMD) [75], potential of mean force (PMF) calculations [76,77], alchemical free energy calculations [78,79], and mostly free energy perturbation (FEP) [80,81,82,83,84], including in silico alanine screening [85,86,87].

The combination of MD simulations with quantum mechanical/molecular mechanical (QM/MM) [88,89] potentials allows researchers to obtain more information about protein functionality [90,91,92]. However, the use of quantum mechanics approaches to assess overall protein stability remains computationally expensive [93,94]; classical MD and QM/MM methods are more commonly used in parallel, rather than in combination.

While the majority of MD simulations are performed for proteins in water solutions in the presence of NaCl (saline concentration), or just counterions added, the effects of the environment are being actively explored. These are physical factors such as temperature [81,95], including processes leading to heat and cold denaturation [96,97], and pressure [98,99,100]. Studies of chemical factors are very extensive: effects of pH, role of hydrogen bonding, salt bridges, and hydration [101,102,103,104]; influence of various cosolvents and additives (osmolytes and crowding agents, including modeling of cytoplasm) [105,106,107,108,109,110,111,112,113,114,115,116]. The improvement of solubility [117,118] and the effect of point mutations on protein–protein interactions, and thus stability of quinary structures [119], are of biotechnological interest.

In addition to an exploration of the effect of point mutations on already folded—experimental and predicted—3D structures, MD simulations are used to address the folding process, in spite of Levinthal’s paradox [114,120,121,122]; unfolding [96,97,123]; and dynamics of intrinsically disordered proteins (IDPs) [82,124]. Such tasks particularly highlight the problem of computational cost and conformational sampling in MD simulations [125]. One of the possible ways to relieve computational costs is to transition from all-atom to coarse-grained models of proteins [102,114,126]. Other options are the use of accelerated molecular dynamics methods [84,127] and enhanced sampling techniques [84,128,129,130,131], including machine learning approaches [132,133,134].

Still, MD simulations require access to supercomputer facilities. This could be an issue for experimental teams. Beyond MD, there is a big family of computationally cheaper, easier-to-master bioinformatics tools, often available as web servers. These are sequence- and structure-based tools for predicting protein stability and the effects of mutations [135,136,137,138,139], proposing mutations for improving stability [140,141], and developing nanoscale protein materials [142]. During past years, FoldX [143] and Rosetta [144] were of particular popularity [145,146,147,148].

Now, we have entered an era of artificial intelligence (AI) technologies. Machine learning (ML) tools based on artificial neural networks (ANNs) for the prediction of the effects of mutations on protein stability and the prediction of protein–protein and protein–ligand interactions are being actively developed. These emerging approaches, algorithms and tools give the impression that we are living through a technological revolution [149], and thus it is impossible at the moment to provide a complete list or name a few most popular or most effective approaches or tools. Some of the most recent papers reporting, comparing, and reviewing AI and ML-based approaches for the prediction of point mutation effect on protein stability are [150,151,152,153,154,155].

The recent release of the AlphaFold2 AI program for protein structure prediction [58,156] is widely seen as a breakthrough in the field of protein structure prediction [157,158]. Though its ability to predict the structural and functional effects of mutations is being debated [159,160,161], AlphaFold2-based approaches are already being developed [162].

## 7. Electrophoresis of Proteins under Denaturing Conditions

There are two ways to consider electrophoresis under denaturing conditions. The first one was popularized in the 1970s and involves the use of SDS-PAGE for the identification of monomeric and oligomeric proteins and the determination of the number and molecular size of subunits, the existence of inter-subunit cross-links (e.g., disulfide bonds), and the purity of preparations. This is the application field of SDS and urea gel electrophoresis. It consists of electrophoretic migration, in a fully denaturing medium, of previously denatured proteins, eventually chemically modified (reduction/alkylation of disulfide bridges, creation of specific cross-links). In addition to the determination of the molecular weight of subunits with good precision, the determination of the number of cysteines [163] and the spatial arrangement of subunits in oligomeric structures is possible [164].

The aim of the second approach is to provide information on the stability of proteins and the mechanism of denaturation. The migration of proteins takes place in a progressively denaturing medium. The change in medium can be the progressive change in concentration of a denaturant or in temperature, either toward heat denaturation or cold denaturation. The gradient of denaturing conditions is perpendicular to the direction of migration, i.e., perpendicular to the electric field. Under such conditions, unfolding transitions can be directly visualized after staining. Conversely, the study of refolding or irreversible denaturation can be performed under non-denaturing conditions after complete denaturation or under conditions allowing the detection of unstable intermediates [30,165]. Technical principles and applications of these approaches were reviewed [166].

### 7.1. Electrophoretic Mobility and Denaturation of Proteins

Electrophoresis allows the separation of macromolecules as a function of their size and charge. In free-flow electrophoresis, the free electrophoretic mobility (*m*_0_) of proteins, regarded as charged spherical particles, is described by the following relationship:(29)$$m_{0}=\frac{Q}{6\pi \eta R}\frac{\chi_{\kappa R}}{\left(1+\kappa R\right)},$$

In Equation (29), *Q* is the protein net charge at the considered pH; *R* is the molecular radius of the protein; and *χ*_(_*κ_R_*_)_ is the Henri function, depending on *R* and *κ*, the Debye–Hückle parameter. *κ* is the reciprocal of the radius of the ionic layer surrounding the protein. This parameter depends on the medium ionic strength, *μ*:(30)$$\kappa ={\left(\frac{8\pi Ne^{2}}{10^{3}\varepsilon kT}\right)}^{\frac{1}{2}}.\;\mu^{\frac{1}{2}},$$
where *N* is the Avogadro number, *e* is the electric charge of an electron, *ε* is the dielectric constant of the medium, *k* is the Boltzmann constant, and *T* is the absolute temperature. In water at 25 °C, *κ* = 0.827 × 10^8^ *μ*^1/2^. When *μ* is high, the ionic atmosphere is dense and shrunken around the protein (*κR*>>1). At low ionic strength, the ionic layer is large (*κR*<<1). When *κR* varies from 0 to ∞, the Henry function varies from 1 to 1.5 and the ratio *Χ*(*κR*)/(1 + *κR*) varies from 0.909 to 0.0015. In practice, the situation is even more complicated due to the distortion of the counterion cloud in the electric field and the deviation of the protein shape from the ideal sphere. Thus, owing to the importance of *μ* in electrophoretic mobility, electrophoresis can only be performed in a narrow range of ionic strength buffers. Therefore, investigation of protein denaturation by extreme pH, high or very low ionic strength, and high concentration of ionized denaturing agents such as guanidinium chloride is not possible using electrophoresis. This may impair the electrophoretic study of the stability of proteins from acidophilic, alkaliphilic, and halophilic microorganisms. Because the electrophoretic mobility depends also on the size, unfolding/refolding transitions can be observed using electrophoresis, particularly in gels acting as molecular sieves such as PAG. The molecular sieve effect of gels mostly depends on the hydrodynamic properties of proteins (size and conformation) and net charge. Changes in electrophoretic mobility allow the detection of protein conformational and structural changes, dissociation, or aggregation in response to changes in external physical conditions or due to the binding of specific molecules.

The molecular sieve effect of PAG and related polymeric matrices on mobility was thoroughly investigated by Rodbard and Chrambach [167,168]. The semi-empirical equation they used (Equation (31)) describes the mobility, *m*, of a protein moving in a polyacrylamide gel of concentration:(31)$$\log_{10}m=\log_{10}m_{10}-K_{R}T.$$
in this equation, used to make Ferguson plots, *m*_0_ is the free electrophoretic mobility, *T* is the polyacrylamide concentration expressed in percentage of monomers, and *K_R_* is the retardation coefficient. *K_R_* depends on the apparent molecular radius of the protein, *R*, and reflects the molecular sieve effect of the gel.
(32)$$K_{R}=c{\left(\bar{R}+r\right)}^{2}.$$

*c* is an empirical factor depending on the buffer (*pH*, composition, ionic strength), temperature, and gel reticulation; *r* is the radius of a gel fiber. The apparent molecular radius *R* is related to the hydrodynamic volume, *V*, of the protein regarded as a sphere and to the molecular mass *M_r_*:(33)$$V=\frac{4}{3}\pi {\bar{R}}^{3}\;\mathrm{and}\;\bar{R}={\left(\frac{3\bar{\nu }\;M_{r}}{4\pi N}\right)}^{1/3},$$
where $\bar{\nu }$ is the partial specific volume and *N* is the Avogadro number.

The unfolding transition is accompanied by an increase in *K_R_* correlated with the increase in hydrodynamic volume, Δ*V*, and diffusion coefficient, *D*. Thus, since *R*>>*r*, from Equations (32) and (33), it follows that
(34)$$\Delta V=\frac{2\pi }{c^{3/2}}\Delta {\left(K_{R}\right)}^{1/2}.$$

This relation can be used to detect volume changes induced by high hydrostatic pressures by using electrophoresis under hydrostatic pressure [169,170]. For this purpose, electrophoreses are performed in capillary gels of increasing concentration in polyacrylamide (%*T*). As seen in Equation (31), *K_R_* can be determined from the slope of Ferguson plots. Thus, when electrophoresis is performed at different pressures, Δ*V* can be determined from the change in *K_R_* with pressure.

Densitometric analysis of stained gels can lead to apparent values of the diffusion coefficient, *D_app_.* The diffusion coefficient, *D*_0_, is related to *R* via the Stokes–Einstein relationship (35):(35)$$D_{0}=\frac{k_{B}T}{6\pi \eta R}.$$
where *k_B_* is the Boltzmann constant, *T* is the absolute temperature, and *η* is the viscosity of the medium. Thus, *D*_app_ as a function of *%T* can be estimated from the variance of electrophoretic bandwidth and extrapolated to *D*_0_ at %*T* = 0 [171]. Dynamic light scattering measurements of protein solutions under high pressure showed that the pressure dependence of *D* reflects pressure-induced conformational changes, resulting from changes in apparent *R* [172].

### 7.2. Electrophoresis under Hydrostatic Pressure

Electrophoresis under high hydrostatic pressure implies the use of heavy equipment. Miniaturized electrophoresis cells are located inside thermostatted massive high-pressure vessels, filled with non-conducting hydraulic fluids (silicone oils, hexane). Vessels are sealed with screw cylinder heads provided with electric connections. High hydrostatic pressure is generated by manual or electric pumps. The applied pressure can reach 6 kbar (600 MPa). The first apparatus for PAGE under high hydrostatic pressure was built by Hawley, who studied the pressure-induced denaturation of chymotrypsinogen A [173,174]. It was possible to perform electrophoretic runs in PAG rods up to 4 kbar (400 MPa). This apparatus equipped with miniaturized electrophoretic cells was also used to perform affinity electrophoresis under high pressure for volume change determination upon protein binding to gel-immobilized ligands [26,170,175]. Later, several types of high-pressure apparatuses were developed either for slab gels [176,177] or capillary gel rods [178]. A review of high-pressure electrophoresis and a description of apparatuses and conditions of use can be found in [170].

Electrophoresis under high pressure in multiple capillary gels of increasing %T can be used first to investigate pressure-induced dissociation/aggregation transitions accompanying denaturation of multimeric proteins [177,178] and hybrid oligomers of paraoxonase-1 [179]. A different system in a narrow-bore glass tube was also developed for 2D electrophoresis to investigate pressure-induced dissociation of oligomeric proteins [180].

PAGE under high pressure can also be used to determine pressure-induced denaturation. Conformational and volume changes (Equation (26)) accompanying protein unfolding transitions can be detected from electrophoretic mobility changes by using Ferguson plot analysis (Equations (31)–(34)) performed using PAGE under increasing pressure. The change in retardation coefficient, KR, with pressure reflects the change in molecular volume (Equations (32) and (34)). In particular, this allows detecting MG transitions (hydrodynamic volume swelling, correlated with an increase in anilino-naphthalene sulfonate binding) that precede unfolding. In the case of human wild-type and mutant cholinesterases, this transition takes place between 1 and 2 kbar (200 MPa) (Figure 5a) [169,181,182]. Neutron scattering measurements of human AChE under high pressure correlated with these pioneering results (Figure 5b) [183].

The presence of 1M sucrose or sorbitol as a protectant abolishes the MG transition of BChE [169] and protects the enzyme against pressure denaturation up to 2.5 kbar [170].

Gel electrophoresis under pressure can be performed in the presence of immobilized ligands. This affinity electrophoresis at varying concentrations of ligands and pressure allow the determination of the apparent volume changes associated with ligand binding [175] and also, as pressure is increased, the detection of the loss of functional activity that precedes pressure-induced unfolding of the protein [26]. This approach can also be used to investigate the effects of solvent/cosolvent that protect or favor pressure unfolding. For example, replacing water with heavy water as the solvent was shown to shift the loss in binding affinity of human BChE for a ligand by 0.4 kbar (40 MPa) (cf. Figure 2A). The loss in binding affinity is correlated with the pressure-induced MG transition of cholinesterases (Figure 5).

### 7.3. Electrophoresis in the Presence of Non-Charged Denaturing Agents

Electrophoresis in multiple gels containing increasing concentrations of non-ionized denaturing agents can provide qualitative information on the unfolding transition [184]. However, the introduction of transverse urea gradient gel electrophoresis (TUGGE) by Creighton allowed direct visualization of unfolding and refolding transitions, detection of transient intermediates, and estimation of ${\Delta G}_{U}^{w}$ [185,186]. In addition, working under isofocusing conditions [187] or at different levels of pH and ionic strength allows the investigation of the dependence of protein (un)folding and ligand binding on these parameters [188].

The principle of transverse urea gradient gel electrophoresis (TUGGE) is simple: the protein sample mixed with a tracking dye (bromophenol blue) is layered along the top of a polyacrylamide slab gel. Then, the protein migrates in a gel of constant porosity (%*T*), containing a linear gradient of denaturing agent (urea from 0 to 8 M) perpendicular to the direction of the electric field. The protein migrates as a continuous zone, and the unfolding transition is detected by the decrease in the protein electrophoretic mobility within a concentration range of the denaturing agent that is more or less large. After the tracking dye has reached the end of the gel, the gel is unmolded and stained. Staining for enzyme activity and/or for proteins can be performed. Methodological principles and practical realizations of gels and electrophoretic runs are described in Creighton’s papers [165,185,189]. Figure 6 shows a typical denaturation curve of a protein (human butyrylcholinesterase tetramer, BChE) in TUGG. By applying this method to a highly purified BChE tetramer irreversibly inhibited by different organophosphates, it was possible to show directly that phosphylation of the enzyme active center dramatically affects the conformational stability of the enzyme [190,191].

#### 7.3.1. Analysis of Denaturation Transitions

Numerous pieces of information can be extracted from electrophoretic unfolding transition curves. In the case of simple transitions N⇌D, both conformations are in equilibrium in the transition zone. Then, if the rates of unfolding (*k*_U_) and refolding (*k*_N_) are fast compared to the electrophoretic migration rate, the unfolding curve is continuous and shows a single inflection point. The apparent rate of transition is *k*_app_ = *k*_U_ +*k*_N_, and its value is minimum at the transition midpoint where *k*_U_ = *k*_N_ [189].

The value of *k*_app_ determines the profile of the transition. Thus, at any point of the transition curve, the electrophoretic mobility (*m*), i.e., the migration distance, is the weighted average of the mobility of the *N* and *U* states. The mobility is, therefore, a measure of the unfolded fraction, *f_U_*.
(36)$$f_{U}=\frac{\left[U\right]}{\left[N\right]+\left[U\right]}.$$

Thus, in the transition zone where concentrations [*N*] and [*U*] are of the same order, *K_U_* and, thus, Δ*G_U_* can be determined.
(37)$$K_{U}=\frac{m_{N}-m}{m-m_{U}}.$$

The comparison of urea concentrations at the transition midpoint [*d*]_0.5_ and *m* values allows the detection of stability differences between iso/allelozymes; chemically modified proteins, e.g., BChE irreversibly inhibited by organophosphorus compounds [190,191]; and engineered mutant proteins [193].

#### 7.3.2. Electrophoretic Profiles of Denaturation Transitions

In the case of small proteins, unfolding transitions are abrupt and do not reveal the presence of intermediates between *N* and *U* states. The examples of ribonuclease, cytochrome c, α- and β-lactoglobulin, lysozyme, staphylococcal nuclease, and protein disulfide isomerase unfolding are typical two-state transitions [189,194,195,196,197]. The presence of disulfide bridges decreases the amplitude of transitions. In addition, the increase in the net charge of proteins during the unfolding process may overcome the decrease in electrophoretic mobility due to the increase in hydrodynamic volume [198].

The unfolding process of proteins composed of several subunits or of multiple domains is more complex. The complexity of unfolding transitions as revealed by TUGGE can be illustrated by several examples: penicillinase [185], α subunit of tryptophane synthetase [199], transferrin [200], phosphorylated human butyrylcholinesterase tetramer [190], and staphylococcal nuclease [196]. Multi-domain proteins unfold sequentially as a function of the stability of each domain. For example, the unfolding of calmodulin (two independent domains) is sequential, showing two consecutive smooth transitions [165]. These multiple conformational states are detected if the rates of their conformational transitions are slow compared to the rate of electrophoretic migration. Thus, unfolding transitions of these proteins can take place within a large concentration range of denaturing agent, reflecting a low cooperativity; they can show several inflection points and even different discontinuous steps. In the case of isoforms and conformers, the transition zone can be split. In the case of slow processes, the transition curve shows a discontinuity. Lastly, the existence of spurs reveals the formation of stable intermediates. In the latter case, a fraction of the protein is unable to rapidly interconvert in the transition zone. This phenomenon reveals the existence of a physical barrier to refolding. Slow isomerization of proline residues may be the cause. Figure 7 summarizes most of the unfolding/refolding transitions as observed by TUGGE.

Finally, TUGGE at different pH levels or isofocusing at different urea concentrations provides important information on the *pH* dependence of the unfolding of oligomeric proteins [201].

#### 7.3.3. Estimation of the Net Stability of Proteins from TUGGE Transition Curves

The net stability of proteins, ${\Delta G}_{N}^{w}$, can be estimated by linear extrapolation of the apparent stability, Δ*G_N_*_app_, to zero concentration of urea [195,198] (Figure 8).

For a simple unfolding transition, assuming, that Δ*G_N_*_app_ (=−Δ*G_U_*_app_) linearly changes as a function of the non-charged denaturing agent, i.e., urea, [*d*], at the midpoint transition where Δ*G_N_* = 0, from Equation (14), it follows that
(38)$$\Delta G_{N}^{w}=m{\left[d\right]}_{0.5}.$$

Then, the change in Δ*G_N_* with [*d*] is expressed by Equation (39):(39)$$\Delta G_{N}=\Delta G_{N}^{w}\left(1-\frac{\left[d\right]}{{\left[d\right]}_{0.5}}\right).$$

The slope of the transition curve is
(40)$$\frac{\partial \Delta G_{N}}{\partial \left[d\right]}=\frac{\Delta G_{N}^{w}}{{\left[d\right]}_{0.5}}.$$

Because the fraction of unfolded protein, *f_U_*, varies between 0 and 1 as a function of [*d*], it can be expressed as a function of Δ*G_N_*:(41)$$f_{U}=\frac{1}{1+K_{N}}=\frac{1}{1+\exp\left(-\frac{\Delta G_{N}}{RT}\right)}.$$

Close to the transition midpoint where *f_U_*,_[*d*]_0.5__ = 0.5, it is assumed that *f_U_* change is approximately linear with [*d*]. Therefore, the slope of the transition curve for *f_U_* vs. [*d*] is
(42)$$\frac{\partial f_{U,{\left[d\right]}_{0.5}}}{\partial \left[d\right]}=\frac{1}{{\left[d\right]}_{0.5}}\frac{\Delta G_{N}^{w}}{4RT}=-\frac{m}{4RT}$$

The two scales, Δ*G_N_* and *f_U_*, are interconnected through the factor 4RT. Because Δ*G_N_*_,[*d*]_0.5__ = 0, the positions of plateaus corresponding to states *N* and *U* provide, on the free energy scale, the values −2*RT* and +2*RT*, respectively (Figure 8). Extrapolation to [*d*] = 0 of the tangents to the transition curve at the transition midpoint gives an estimation of ${\Delta G}_{N}^{w}$ on the *xRT* axis.

For small proteins such as ferricytochrome c, the estimated value of ${\Delta G}_{N}^{w}$ (41.9 kJ·mol^−1^) is of the order of magnitude of values obtained by DSC (33.4 kJ·mol^−1^) or guanidinium chloride (30.5 kJ·mol^−1^) denaturation [165]. Studies on staphylococcal nuclease also confirmed the good correlation between ${\Delta G}_{N}^{w}$ determined by DSC and TUGGE [202]. However, in the case of multi-domain large proteins and oligomers, TUGGE may lead to underestimation of ${\Delta G}_{N}^{w}$ [190]. In these cases, estimated apparent values of ${\Delta G}_{N}^{w}$ are useful indexes for comparing the stability of mutated or chemically modified proteins [191].

#### 7.3.4. Other Denaturing Agents for TdGGE

Urea has a certain number of limitations. The main drawback is the progressive formation of cyanate ions in urea solutions. Cyanate ions may carbamylate primary amines and thiol groups in proteins in alkaline buffers at a temperature higher than 35 °C. In addition, 8 M solutions of urea are the solubility limit and do not allow working at low temperature. Thus, other non-ionic denaturing agents have been investigated. These can be non-polar solvents that weaken hydrophobic interactions or polar solvents, acting like urea by competing with intramolecular H-bonds. However, this denaturation is in general less complete than that with urea. Other compounds are alkylureas, denaturing agents more potent than guanidine hydrochloride, and propylene carbonate; these compounds are more stable than urea and do not react with amino acid side chains [203,204]. Some of these agents may inhibit polyacrylamide polymerization or lead to gels of heterogeneous porosity [205]. Among them, tetramethylurea, a liquid of F = −1.2 °C, was successfully used in TdGGE from 0 to 2.5 M as a urea substitute [206]. Sulfolane was also used in polyacrylamide-acryloylmorpholine gels up to 8 M. In these gels, the unfolding transition of hydrophilic proteins takes place at concentrations lower than those with urea. However, circular dichroism spectra suggest that sulfolane denaturation is less extensive than that with urea [207]. Finally, recent results provided evidence that urea is the best non-ionic denaturing agent so far, at least for *m*-based determination of ${\Delta G}_{N}^{w}$ [42].

### 7.4. Electrophoresis at Different Temperatures

Electrophoresis at different temperatures can be performed in gel rods or plates at a constant temperature above or below 0 °C and can provide qualitative information on heat- or cold-induced unfolding transitions. As for denaturation by non-charged chemical denaturing agents, the most informative and appealing approaches are electrophoreses in transverse temperature gradient gels (TTGGE) to investigate heat-induced unfolding or cold-induced unfolding. Moreover, this technique also allows the investigation of the influence of a solvent or a neutral agent, e.g., urea, on thermal stability.

#### 7.4.1. Heat Unfolding Study by Transverse Temperature Gradient Gel Electrophoresis

TTGGE is conceptually similar to TUGGE. It was introduced by Thatcher to monitor conformational changes accompanying heat denaturation of proteins and nucleic acids and to determine half-transition temperatures [208]. Initially applied to the study of the relative thermal stability of *Drosophila* alcohol dehydrogenase and lactico-dehydrogenase allozymes, it allowed the detection of transient intermediates of denaturation and irreversibly denatured forms [208,209].

The first apparatus designed by Thatcher was a vertical slab gel apparatus capable of operating in a temperature range of 10 to 60 °C. The gel plate is mounted between two aluminum blocks connected at their ends to two thermostats set to temperatures *T*_min_ and *T*_max_. The temperature gradient is established between the two ends by simple thermal conduction. The gradient linearity can be controlled by thermistors introduced in wells drilled into the aluminum plates. Electrophoresis buffers are chosen so that pH change with temperature does not exceed 0.3 units over the temperature range. The reproducibility of runs and transition curves on this apparatus was excellent, providing mid-transition temperature, *T*_0.5_, with an accuracy of ±1 °C. Another system based on a commercial flat-bed electrophoresis apparatus was also described [210]. In this apparatus, the slab gel is placed on a horizontal copper block connected as in the previous apparatus to two thermostats. This system is operative in a larger temperature range, between 10 and 80 °C [211].

TTGGE can be used to study the heat-induced unfolding of proteins and nucleic acids and the dissociation of protein–nucleic acid complexes. As for TUGGE, analysis of transition curves, continuous or discontinuous, showing spurs provide insight into the mechanisms of thermal unfolding [212,213]. It can be reversely used to study the folding of heated proteins.

#### 7.4.2. TTGGE to Study Cold Denaturation of Proteins

Cold denaturation of proteins is a reversible process with thermodynamic characteristics opposite to heat denaturation, i.e., heat release and entropy decrease. In addition, unlike heat denaturation, no chemical alterations are produced. The electrophoretic study of cold denaturation implies operating in media containing antifreeze agents in specially thermostatted apparatuses or in cold rooms. Antifreezes added to electrophoresis buffers can be ethylene glycol (50% *v*/*v*) [47], dimethylsulfoxide (50% *v*/*v*) [214], or ethylene glycol/methanol mixtures [215]. The presence of cosolvents may dramatically affect protein stability, either in favoring their denaturation or in protecting their structure [216]. Down to −10 °C, electrophoresis can be performed in standard polyacrylamide gels [47]. However, below −10 °C, polyacrylamide gels become opaque, likely due to the vitreous transition of polyacrylamide. For working at lower temperatures, ternary copolymers of acrylamide-ethylacrylate-bismethylene acrylamide stable down to −40 °C can be used. The temperature gradient and the of monitoring its linearity along the gel can be controlled using a thermistor. A tracking dye (pyronin Y) can be used to monitor the migration. Initially, subzero electrophoresis was applied to the study of dissociation of oligomeric proteins, antigen–antibody complexes, and hybrid proteins [217,218]. Slab gel electrophoresis in gels of T = 8%, C = 1.85% acrylamide-ethylacrylate-bisacrylamide in a transverse gradient of subzero temperature was applied to investigate protein cold denaturation (Figure 9) [219].

Although subzero TTGGE of proteins is very sensitive in detecting subtle reversible conformational changes that correlate with cold-induced changes observed in circular dichroism spectra, multiple difficulties in practical realization restrain the use of this method.

#### 7.4.3. Capillary Zone Electrophoresis at Different Temperatures

Progress in the electrophoretic investigation of protein stability was accomplished due to the development of capillary zone electrophoresis (CZE) at different temperatures, up to 91 °C, in a 50 μm diameter fused-silica capillary of 87 cm length by Rochu et al. [220]. The authors used CZE for the investigation of folding, unfolding, and refolding of proteins, mostly cholinesterases and phosphotriesterases. A review by Righetti and Verzola [4] highlighted the potential of this approach. Later, the theoretical background was formalized by Gavina and Britz-McKibbin [5]. To our knowledge, CZE has not yet been used to study the cold denaturation of proteins. The reason is likely related to the use of antifreeze cryoprotectants that would impair cold-induced unfolding transitions.

Electrophoretic mobility and calculation of thermodynamic parameters of unfolding are determined from apparent migration times of proteins and areas under the protein electropherograms:(43)$$\mu =\mu_{app}-\mu_{eo}=\frac{L_{t}L_{d}}{V}\left(\frac{1}{t_{app}}-\frac{1}{t_{eo}}\right).$$
where *μ_app_* and *μ_eo_* are the apparent and electroosmotic mobilities (cm^2^ V^−1^ s^−1^), respectively. *L_t_* is the total length of the capillary, and *L_d_* is the capillary length from the inlet to the diode array detector (absorbance λ ranging between 195 and 300 nm). *V* stands for the applied voltage (10 kV), *t_app_* is the apparent migration time, and *t_eo_* is the migration time of the electroendosmosis flow marker (N,N-dimethylformamide).

Van’t Hoff plots for protein unfolding are linear in the transition region, allowing determination of enthalpy changes of unfolding (Δ*H_m_*) at the midpoint transition temperature (*T_m_*), and Δ*C_p_* is calculated from the Kirchhoff equation (Equation (25)). Then, thermodynamic parameters are determined according to the Gibbs–Helmholtz equation (Equation (44)):(44)$$\Delta G_{T\;}=\Delta H_{m}\left(1-\frac{T}{T_{m}}\right)-\Delta C_{p}\left[\left(T-T_{m}\right)+TLn\left(\frac{T}{T_{m}}\right)\right].$$

As seen, the CZE approach allows access to Δ*G_N_* vs. *T* diagrams (Figure 10) [221,222,223]. Thermodynamic parameters for the unfolding of different proteins determined by CZE are in agreement with parameters determined by DSC [220,224,225,226].

### 7.5. Electrophoresis after Exposure to Extreme Physical Conditions

Electrophoresis of denatured proteins provides information on the reversibility of unfolding transitions, chemical modifications caused by denaturing conditions, degree of denaturation and subsequent events, refolding, misfolding, and conformational drifts.

#### 7.5.1. Analysis of Renaturation Processes

Renaturation of unfolded proteins can also be investigated by TUGGE or TTGGE [228] or under non-denaturing conditions [229]. By operating at low temperatures or by making specific amino acid group chemical modifications, it is possible to trap and identify transient intermediates.

With certain proteins, the unfolding process may continue after the application of the denaturation stress has ended (e.g., after pressure release, return to standard temperature conditions, dilution of denaturing agents). Slow return to the *N* state or slow transitions to different conformational states can be observed. These phenomena are related to hysteresis and conformational drift [230]. Conversely, continuations of the denaturation process after a return to standard conditions can be observed, such as the so-called remnant heat inactivation process [231].

#### 7.5.2. Direct Observation of Unfolding Reversibility

After the interruption of heat treatment or release of high hydrostatic pressure, refolding and/or reassociation of monomers in oligomeric structures can be observed by electrophoresis under standard conditions. Chemical modifications (e.g., by glutaraldehyde) of denatured proteins and separation of reaction products by SDS-PAGE provide information on the extent of unfolding [232].

#### 7.5.3. Evidence for Irreversible Denaturation

Identification of a stable denatured (*D*) state, dissociated oligomers, or irreversible aggregates of unfolded proteins after single or combined action of heat, pressure, ultrasound, chemical, and solvent denaturation can be performed by PAGE under standard conditions [203,233,234,235,236]. Then, protein staining, immunoblotting, or affino-immuno-electrophoresis reveals the threshold of enzyme dissociation/inactivation or loss of immunoreactivity.

This approach can be used to compare genetic variants [237,238] or to detect discrete deteriorations in a “homogeneous protein” [233] (Figure 11), and to quantify the effect of a specific chemical modification on enzyme stability. The latter was in particular applied to analyze the stability of different phosphorylated forms of BChE submitted to the combined action of pressure and an organic solvent (S = propylene carbonate). Phenomenological analysis of pressure–solvent (*d*) denaturation maps at the midpoint of the denaturation process provides information on denaturation steps and gradual stability change of the chemically modified (phosphorylated) enzyme. The d[*d*]/d*P* curves represent contours of constant molar fraction (*f*_D_) of denatured state on the pressure–solvent plane.
(45)$$\frac{d\left[d\right]}{dP}=\frac{\Delta V_{D}^{\neq }}{m^{\neq }}.$$

At the midpoint of denaturation, *f*_D_ = *f*_N_, the area under the curves (d[*d*]/d*P*)_1/2_ reflects the relative stabilizing energy of the chemically modified enzyme [203]. This approach can be applied to the comparative study of the stability of allozymes, muteins, and extremozymes.

## 8. Conclusions

The electrophoretic techniques (PAGE, CZE) operating under denaturing conditions or in standard conditions on post-denatured proteins have become complementary to the most sophisticated biophysical techniques and in silico computational chemistry approaches for investigating protein stability, unfolding/refolding processes, and irreversible physicochemical damages in protein structures. Electrophoretic techniques are simple, fast, and very sensitive, and they provide direct and pictural information on the energetics of unfolding (and refolding) transitions and about genetic, structural, and physicochemical factors that determine the stability of macromolecular scaffolds and oligomeric structures. In particular, on a single electrophoretic gel, they allow stability comparisons between natural and engineered proteins and between iso- and allo-proteins. Moreover, under certain conditions, electrophoretic techniques are capable of detecting intermediate states of unfolding/refolding such as molten globules, and slow/rare events such as conformational drifts occurring after the release of denaturing conditions. Thus, these techniques mostly developed between the end of the past century and the first decade of the present century are still valuable tools. They provide important structural and functional information about natural and engineered proteins, particularly extremozymes of industrial and biotechnological interest; food processing; and the stability of biopharmaceuticals, including nanoformulations of enzymes of medical interest.

## Figures and Tables

**Figure 1 molecules-27-06861-f001:**
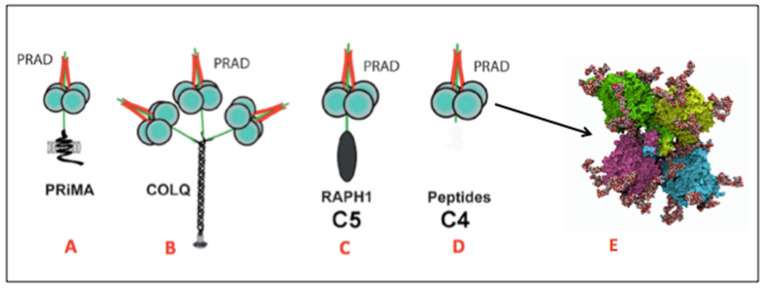
Quaternary and quinary structure of cholinesterases. Both ChEs display the same monomeric fold (α/β) and similar upper 3D structure levels. (**A**) Membrane-anchored acetylcholinesterases (AChE) anchored by PRiMA (proline-rich membrane anchor) in central nervous system cholinergic synapses. (**B**) Matrix-anchored cholinesterases anchored by ColQ (collagenous subunit) at neuromuscular junctions. (**C**) Soluble human plasma butyrylcholinesterase C_5_ isoenzyme: tetrameric form linked to Raph 1 subunit [9]. (**D**) Soluble human plasma butyrylcholinesterase (BChE) tetrameric form: dimer of disulfide-bonded dimers linked through PRAD (proline-rich attachment domain) peptide. (**E**) Three-dimensional structure of human BChE tetramer as determined by cryo-electron microscopy [10,11]. Nine asparagine-linked glycan chains of complex type per subunit correspond to 24% of the total protein mass (340 kDa).

**Figure 2 molecules-27-06861-f002:**
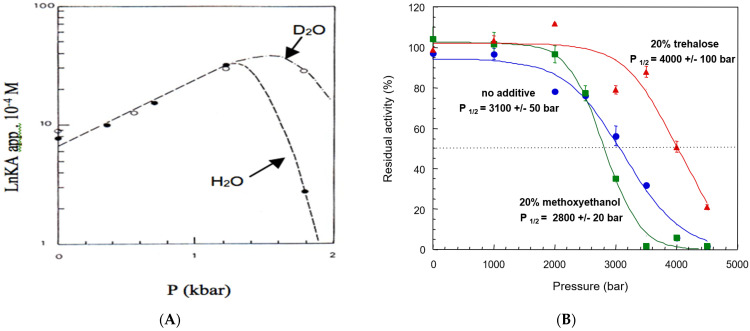
Effect of solvent, cosolvent, and additive on binding affinity (**A**) and catalytic activity (**B**) of human BChE submitted to high pressure. (**A**) Protective effect of D_2_O on pressure-induced loss of affinity for a ligand of human BChE (phenyl-trimethylammonium) in Tris/Gly buffer, pH 8.4 in water or pH 8.8 in 90% D_2_O at 35 °C (redrawn from [26]). (**B**) Loss in activity of human BChE in 10 mM Tris/HCl, pH 7.4 at 25 °C, as a function of hydrostatic pressure in the absence of a cosolvent (blue curve), in the presence of a stabilizer additive (red curve), and in the presence of a destabilizing cosolvent (green curve). Cosolvent (20% *v*/*v*) and stabilizer (20% *w*/*v*) lead to shifts in *P*_1/2_, the pressure causing 50% of enzyme inactivation (unpublished).

**Figure 3 molecules-27-06861-f003:**
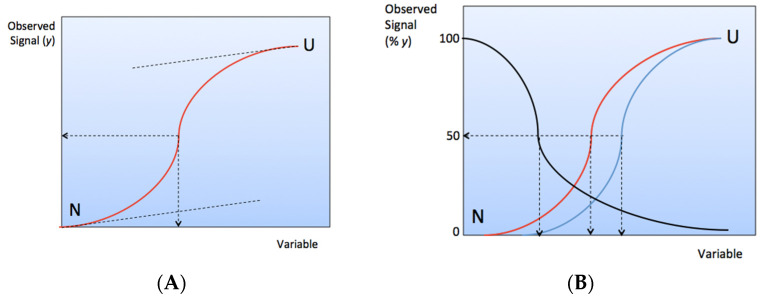
(**A**) Schematic diagram of reversible protein unfolding transition. At the transition midpoint [*N*] = [*U*] and *K_U_* = 1. (**B**) Multistep unfolding transition of an enzyme. Loss of enzyme activity (black curve) takes place first. Then, as the intensity of the perturbing variable is increased, buried thiol groups (cysteine residues) may become exposed to solvent (red curve), and thiol groups may react with chromogenic reagents such as dithiobisnitrobenzoic acid (DTNB). At higher variable intensity, where more extended conformational changes occur, tryptophan residues emerge on the protein surface and can be detected by the change in protein fluorescence (blue curve). Secondary structure changes can be observed at higher variable intensity by means of Raman and FT-IR spectroscopies.

**Figure 4 molecules-27-06861-f004:**
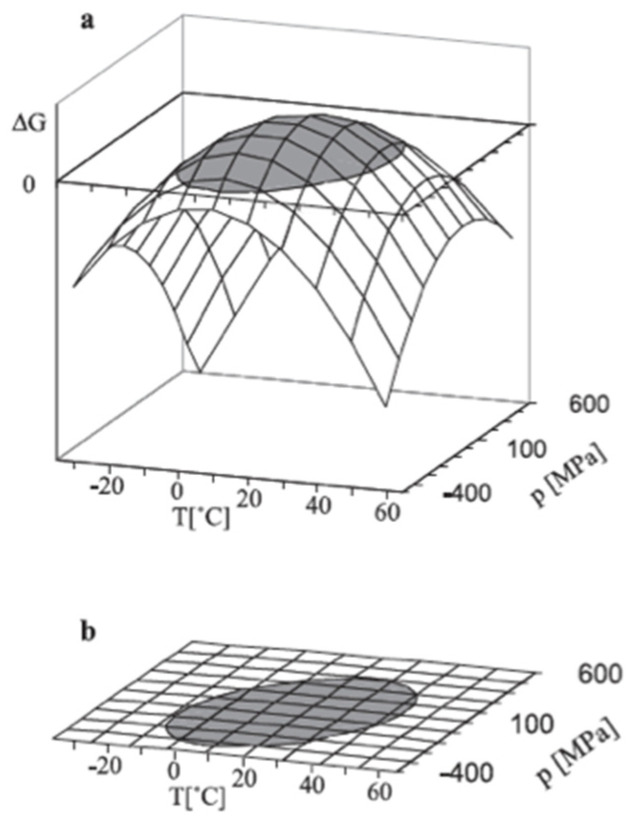
Pressure/temperature reversible denaturation of chymotrypsinogen. (**a**) Δ*G*_U_ as a function of temperature and pressure; (**b**) elliptical phase diagram as the horizontal projection of Δ*G*_U_ = *f*(*P*,*T*) for Δ*G*_U_ = 1, the grey area shows the region where the native protein is more stable (reproduced from [44] with permission from Elsevier).

**Figure 5 molecules-27-06861-f005:**
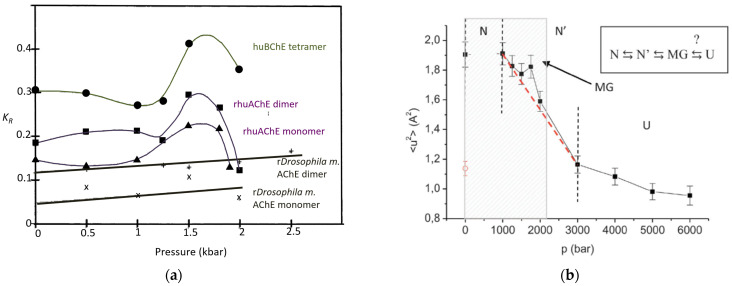
Pressure-induced molten globule (MG) transition of cholinesterases. (**a**) PAGE under high pressure: change in the retardation coefficient (*K_R_*) of different cholinesterases with high pressure in 8.26 mM Tris/Gly pH 8.3 at 10 °C (redrawn from [181]; (λ) human BChE tetramer; (ν) human AChE dimer; (▲) human AChE monomer; (✕) recombinant *Drosophila melanogaster* AChE dimer; (+) recombinant *Drosophila melanogaster* AChE monomer). Although all ChEs have the same fold, unlike human ChEs, the *Drosophila* enzyme that has low sequence homology with human enzymes does not show the MG transition around 1.5 kbar. (**b**) Neutron scattering of human AChE (reproduced from [183] with permission from the Royal Society of Chemistry). Mean square displacement (MSD) as a function of pressure, the MG transition is at 1.75 kbar.

**Figure 6 molecules-27-06861-f006:**
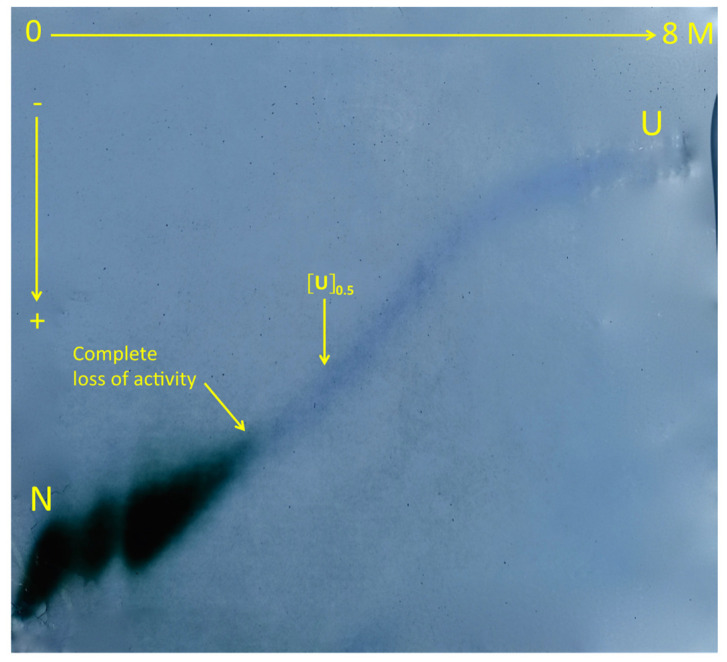
TUGGE of human butyrylcholinesterase tetramer. BChE (0.5 mg highly purified protein (>90% pure) in 0.5 mL electrophoresis buffer) was layered along the top of the 0–8 M TUGG. Migration was performed at 0 °C under the constant intensity of 30 mA for 5.5 h in 50 mM Tris/Gly buffer pH 8.4. The unfolding transition curve was revealed by double staining: after electrophoresis, the gel was first stained for activity according to Juul’s method using butyrylthiocholine iodide (1 mM) as the substrate [192]. This led to a green zone of enzyme activity. Then, the protein was stained with Coomassie Brilliant Blue. Activity staining revealed that the enzyme completely loses its activity at 2.9 M urea, and protein staining showed a continuous unfolding transition curve with the midpoint at [U]_0.5_ = 3.5 M urea (unpublished, see [190] for methodology details).

**Figure 7 molecules-27-06861-f007:**
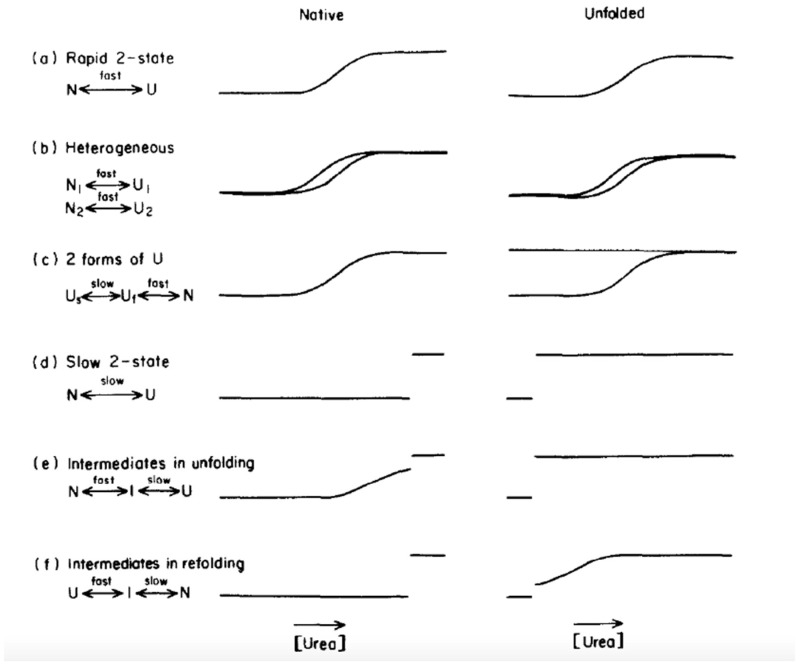
TUGGE patterns for most unfolding and refolding transitions. Left panel: unfolding transitions of native proteins; right panel: refolding transitions of the same proteins previously fully unfolded and applied to the top of the gel (reproduced from [185] with permission from Elsevier). (**a**) Unfolding and refolding processes obey the two-state model: rapid transition between N and U states; (**b**) the native state (N) is heterogeneous, each population unfolds and refolds rapidly at different denaturant concentrations; (**c**) two slowly interconverted unfolded states exist, one refolds rapidly, the second one refolds slowly; (**d**) unfolding and refolding are slow processes, so the initial state, N or U, persists in denaturing/non-denaturing conditions; (**e**) rapid and reversible partial unfolding produces a stable intermediate (I) prior to complete unfolding; (**f**) this case is opposite to (**e**), with the production of a stable refolding intermediate (I).

**Figure 8 molecules-27-06861-f008:**
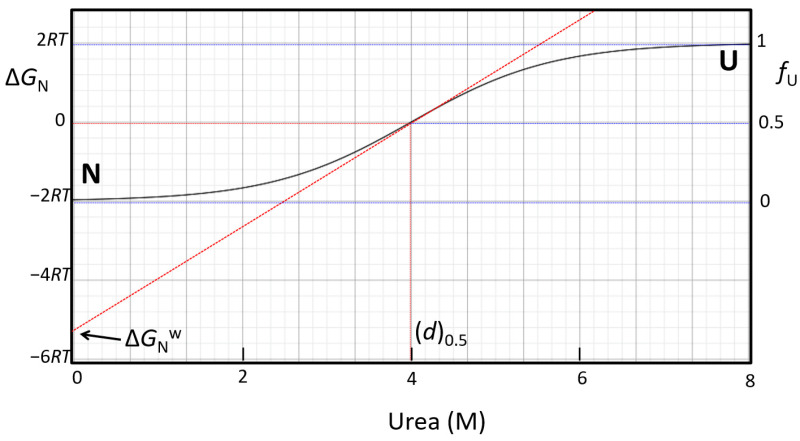
Estimation of the net stability of proteins, ${\Delta G}_{N}^{w}$, by linear extrapolation to zero concentration of denaturing agent, *d*.

**Figure 9 molecules-27-06861-f009:**
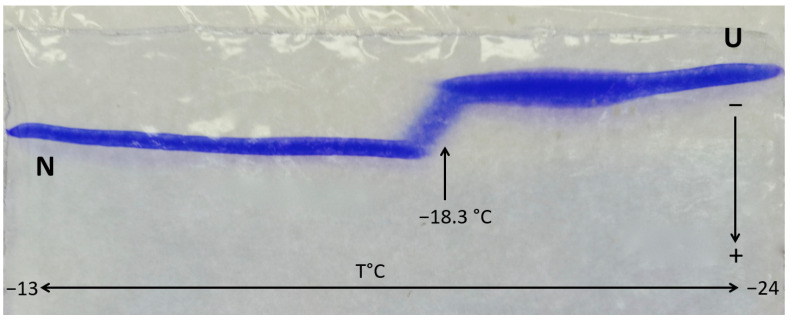
Subzero TTGGE of RNase A (0.68 mg/mL) between −13° and −24 °C. Electrophoresis was carried out for 3 days under 1800 V and 3 mA. The applied electric field was 130 V/cm. The protein was stained with Coomassie Brilliant Blue (for technical details, see [219]). The discontinuity of the transition curve around −18.4 °C reflects a slow transition between the *N* and *U* states as predicted when the rates of unfolding and refolding are slow [186].

**Figure 10 molecules-27-06861-f010:**
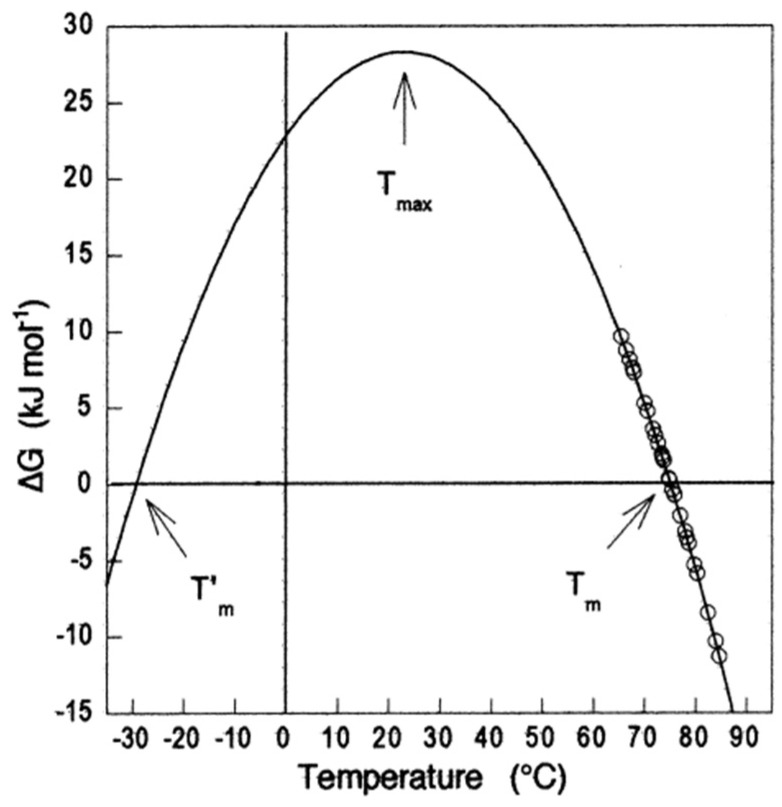
Stability curve, Δ*G_N_* vs. *T*, of β-lactoglobulin B from CZE at different temperatures in 100 mM sodium phosphate, pH = 6.2. T’m and Tm are the transition temperatures corresponding to cold- and heat-induced unfolding. Data were fitted according to the Gibbs–Helmholtz equation (Equation (44)). Reproduced from [227] with permission from Elsevier.

**Figure 11 molecules-27-06861-f011:**
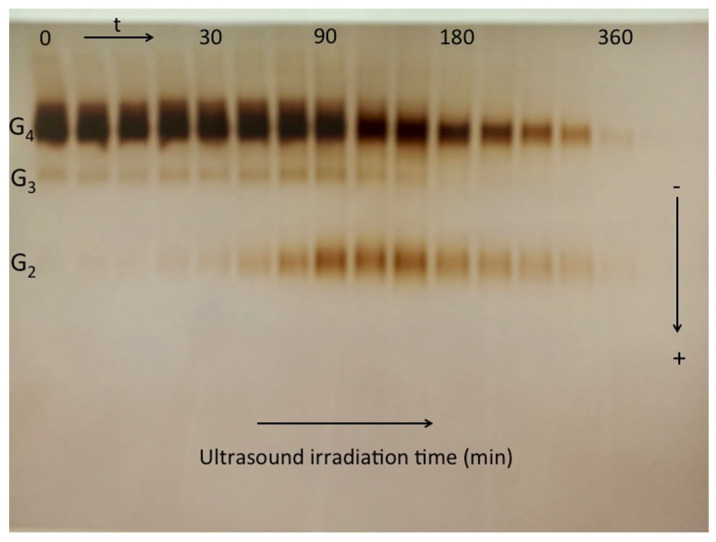
Slab gel PAGE under non-denaturing conditions of human BChE tetramer (G_4_ form) following exposure to pulsed ultrasound (85 W) for up to 360 min. The enzyme was revealed by activity staining. The preparation contained 90% of tetramer and less than 10% trimer (G_3_). Under ultrasound irradiation, a fraction of tetramer dissociated progressively into active “nicked” trimer and dimer (G_2_) that were slowly inactivated. The dimer is formed of disulfide-bonded monomers (G_1_). However, G_1_ was not transiently observed because the “nicked” monomeric subunits are inactive (unpublished data; for technical details, see [233]).

## Data Availability

Not applicable.

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
