# Peer review of "Conformational Stability and Denaturation Processes of Proteins Investigated by Electrophoresis under Extreme Conditions"

_molecules, 2022, doi:10.3390/molecules27206861_

Round 1
Reviewer 1 Report
In this review, the authors presented an overall picture on electrophoretic techniques that are useful to provide information on energetics on folding and to establish the physicochemical factors that influence protein stability. However, several concerns exist there in the manuscript. I would like to recommend the manuscript to be published in Molecules after minor revisions. Detailed comments are appended as below.
Too many equations are offered in the manuscript, especially in 6.1. We recommend improving Figure 8 clarity (text: rapid 2 state etc) and to combine Figures 3 and 4 (if possible). Could you have a similar high for the plots A and B from Figure 2? or it is feasible to redesign (redraw) panel A?
Author Response
We understand that the manuscript contains many equations, and in particular in section 6.1. However, most biochemists and electrophoresis practionners are not familiar with concepts and theory of protein stability/unfolding, and it was mandatory to provide a minimum background in physical chemistry, at least to understand the meaning of the graphical abstract and information that can be extracted from electrophoretic transition curves.
Figures 3 and 4 were combined (now Fig. 3A and B). Thus, all figures were re-numbered.
Figure 2 was modified as requested
Legend of Figure 8 (now Fig. 7) was expanded to clarify the different transition curves. It is now understandable. Thanks for your recommendation.
Reviewer 2 Report
I think that the paper is written too extensively for a reviuw article. This is more like a book. Too much theory is presented and the essence is lost, which is the use and way in which the results obtained by electrophoretic techniques provide useful information about the protein. Details of possible examples and their practic use are missing. The first 6 chapters are too extensive for a review paper. And from 7 onwards are very useful and interesting but misrepresented because the descriptions of the techniques are missing, they are just somehow enumerated with a quote. My advice is to adapt this text for a book and for this purpose write a new abbreviated one that only refers to electrophoretic techniques and their usefulness in practice.
Author Response
We thank Reviewer 2 for his efforts to read a manuscript containing a lot of theory and equations. However, in such a feature article about electrophoresis under extreme physico-chemical conditions where the main goal is to access stability parameters of proteins, it’s impossible to avoid theory. Without basic equations, references to theory of protein stability, and effects of physical pramaters such as temperature (heat and cold) and hydrostatic pressure on native protein structures, it would be impossible to understand how from electrophotetic mobility changes we can access important thermodynamic quantities (stability), and get information on mechanisms of unfolding (and refolding).
The manuscript is illustrated with many exemples and numerous references of proteins submitted to electrophoresis (in polyacryamide gels and in capillary tubes) in the presence of denaturing agents, high pressure, subzero temperatures and heat.
The experimental techniques are well described in quoted articles, and their description would have considerably increased the size of the manuscript.
The manuscript has 25 pages, to adapt to text to a book would imply a huge expansion. This is far beyond the goal of this article. It could be a future project…
On the other hand, to reduce the size of the article, would make very difficult to understand how heat, pressure, organic solvent and other perturbing variables act on proteins, affect electrophoretic mobility and leads to free energy of stability.
Round 2
Reviewer 2 Report
OK, you convinced me with your answers!
but still, consider publishing this in a book as well, it would be useful to make it available to biochemistry students as a useful study reference
protein electrophoresis has been unjustifiably neglected in recent years
please understand that my idea was to make your work as visible as possible
Author Response
Thanks for your last comments. I agree with you that electrophoresis has been neglected in the past years. We hope that our review will open eyes to young scientists who did not work in the glory years of SDS_PAGE, Ferguson plots, PAGE and CZE under denaturing conditions.
Sincerely,
Patrick Masson